# A systematic review of clinical health conditions predicted by machine learning diagnostic and prognostic models trained or validated using real-world primary health care data

Hebatullah Abdulazeem[1]*, Sera Whitelaw[2], Gunther Schauberger[1], Stefanie J. Klug[1]

**1** Chair of Epidemiology, Department of Sport and Health Sciences, Technical University of Munich (TUM), Munich, Germany, **2** Faculty of Medicine and Health Sciences, McGill University, Montreal, Quebec, Canada

* hebatullah.abdulazeem@tum.de

## Abstract

With the advances in technology and data science, machine learning (ML) is being rapidly adopted by the health care sector. However, there is a lack of literature addressing the health conditions targeted by the ML prediction models within primary health care (PHC) to date. To fill this gap in knowledge, we conducted a systematic review following the PRISMA guidelines to identify health conditions targeted by ML in PHC. We searched the Cochrane Library, Web of Science, PubMed, Elsevier, BioRxiv, Association of Computing Machinery (ACM), and IEEE Xplore databases for studies published from January 1990 to January 2022. We included primary studies addressing ML diagnostic or prognostic predictive models that were supplied completely or partially by real-world PHC data. Studies selection, data extraction, and risk of bias assessment using the prediction model study risk of bias assessment tool were performed by two investigators. Health conditions were categorized according to international classification of diseases (ICD-10). Extracted data were analyzed quantitatively. We identified 106 studies investigating 42 health conditions. These studies included 207 ML prediction models supplied by the PHC data of 24.2 million participants from 19 countries. We found that 92.4% of the studies were retrospective and 77.3% of the studies reported diagnostic predictive ML models. A majority (76.4%) of all the studies were for models' development without conducting external validation. Risk of bias assessment revealed that 90.8% of the studies were of high or unclear risk of bias. The most frequently reported health conditions were diabetes mellitus (19.8%) and Alzheimer's disease (11.3%). Our study provides a summary on the presently available ML prediction models within PHC. We draw the attention of digital health policy makers, ML models developer, and health care professionals for more future interdisciplinary research collaboration in this regard.

**Editor:** Ágnes Vathy-Fogarassy, University of Pannonia: Pannon Egyetem, HUNGARY

**Data Availability Statement:** All relevant data are within the paper and its Supporting information files.

**Funding:** The authors received no specific funding for this work.

**Competing interests:** The authors have declared that no competing interests exist.

## Introduction

Primary health care (PHC) is considered the gatekeeper, where health education and promotion are provided, non-life-threatening health conditions are diagnosed and treated, and chronic diseases are managed [1]. This form of health maintenance, which aims to provide constant access to high-quality care and comprehensive services, is defined and called for by the World Health Organization (WHO) global vision for PHC [2]. Clinicians' skills and experience and the further continuing professional development are fundamental to achieve these PHC aims [3]. Additional health care improvement can be achieved by capitalizing on digital health and AI technologies.

With the high number of patients visiting PHC and the emergence of electronic health records, substantial amounts of data are generated on daily basis. A wide spectrum of data analytics exist to utilize such data; however, meaningful interpretation of large complicated data may not be adequately handled by traditional data analytics [4]. Tools that could more accurately predict diseases incidence and progression and offer advice on adequate treatment could improve the decision-making process. Machine Learning (ML), a subtype of Artificial Intelligence (AI), provides methods to productively mine this large amount of data such as predictive models that potentially forecast and predict diseases occurrence and progression [5]. The variety of ML prediction models' characteristics provide broader opportunities to support the healthcare practice.

Integrating PHC with updated technologies allows for the coordination of numerous disciplines and views. Integrating PHC with such technologies allows for improvements in health care, which may include patient care outcomes and productivity and efficiency within health care facilities [5, 6]. ML models have been developed in health research–most significantly in the last decade—to predict the incidence of diabetes, cancers, and recently COVID-19 pandemic related illness from health records [7]. A systematic overview of 35 studies published in 2021 investigated the existing literature of AI/ML, but exclusively in relation to WHO indicators [8]. Other literature and scoping reviews examined AI/ML in relation to certain health conditions, such as HIV [9], hypertension [10], and diabetes [11]. Other systematic reviews targeted specific health conditions across multiple health sectors, such as pregnancy care [12], melanoma [13], stroke [14], and diabetes [15]. However, reviews investigating PHC specifically have been fewer [16, 17]. It has been reported that research on ML for PHC stands at an early stage of maturity [17]. Similar to ours, a recently published protocol of a systematic review addressing the performance of ML prediction models in multiple different medical fields was published [18]. However, this protocol does not focus specifically on primary care and its search is limited to the years 2018 and 2019. Hence, the current literature is not enough to identify what diseases are targeted by ML prediction models within real-world PHC. Furthermore, literature investigating the validity and the potential impact of such models are not abundant. To direct the focus toward this gap, we conducted this systematic review to encompass the health conditions predicted through using ML models within PHC settings.

## Materials and methods

We conducted a systematic review in accordance with the Preferred Reporting Items for Systematic Reviews and Meta-Analyses (PRISMA) [19] and the CHecklist for critical Appraisal and data extraction for systematic Reviews of prediction Modelling Studies (CHARMS) [20]. The protocol for our review was registered on PROSPERO CRD42021264582 [21].

### Search strategy and selection criteria

A comprehensive and systematic search was performed covering multidisciplinary databases: 1. Cochrane Library, 2. Elsevier (including ScienceDirect, Scopus, and Embase), 3. PubMed, 4.

Web of Science (including nine databases), 5. BioRxiv and MedRxiv, 6. Association for Computer Machinery (ACM) Digital Library, and 7. Institute of Electrical and Electronics Engineers (IEEE) Xplore Digital Library.

To find potentially relevant studies, we searched literature with the last updated search on January 4, 2022, back to January 1, 1990. The utilized search terms included "machine learning", "artificial intelligence", "deep learning", and "primary health care". Boolean operators and symbols were adapted to each literature database. Hand searches of citations of relevant reviews and a cross-reference check of the retrieved articles were also performed. Conference abstracts and gray literature searches were conducted using the available features of some databases. The full search strategy for all the electronic databases is presented in S1 File. A reference management software (EndNote X9) was used to import references and to remove duplicates.

The inclusion criteria were as follows: primary research articles (peer-reviewed, preprint, or abstract) without language restriction, studies reporting AI, DL or ML prediction models for any health condition within PHC settings, and using real-world PHC data, either exclusive or linked to other health care data. We directed our focus toward these supervised ML models (random forest, support vector machine (SVM), boosting models, decision tree, naïve bias, least absolute shrinkage and selection operator (LASSO), and k-nearest neighbors) and the neural networks.

## Literature screening, data collection and statistical analysis

Title and abstract screening for all records were conducted independently by two researchers through the Rayyan platform [22]. Discrepancies were resolved by discussion. All studies that met the eligibility criteria were included in the systematic review. The process of data extraction was performed by two authors. Items and definitions of extracted data is presented in Table 1.

Health conditions extracted were categorized according to the International Classification of Diseases (ICD)-10 version 2019 [23]. This coding system was selected because it is applied

**Table 1. Items and definitions for data extraction.**

| Item Extracted | Definition |
|---|---|
| Meta-data | First author and year of publication |
| Study Type | According to CHARMS guidelines [20], the types of a prediction modelling studies are:<br>    1. model development without external validation,<br>    2. model development with external validation, or<br>    3. external validation of a predeveloped model with or without model update.<br><br>The included studies are presented in the Results section in three tables categorized according to these three types. |
| Study design | Design of the included studies. |
| Models purpose | 1. Incident diagnostic (occurrence probability of a disorder),<br>2. Prevalent diagnostic (identifying overlooked cases), or<br>3. Prognostic (occurrence probability of a future event). |
| Country | Countries, from which health data were collected to train, test, or validate the models. |
| Source of data | This represents the source of the health data used to train, test, or validate the model.<br>    1. PHC (Data exclusively originated from a PHC settings)<br>    2. Linked data (PHC data linked to other data sources, such as secondary or tertiary health care) |
| Sample size | Number of the population, whom health data were used to train, test, or validate the models. |
| Time span of data | Time period, in which the health care data used for modelling were originally available in the health care system. |
| Health condition | Health condition addressed in the included studies. |

by at least 120 countries across the globe [24]. Considering the countries that apply different coding systems, we used the explicit names of the health condition mentioned in the included studies included to match them to the closest ICD-10 codes.

Descriptive statistics of the extracted data was calculated. The overall number of populations was calculated with considering the potential overlap between the included datasets. This overlap assessment was contemplated based on similarity of source of data, time span of data within each included study, the targeted health condition and the inclusion and exclusion criteria of the participants. The quantitative results were calculated using Microsoft Excel.

### Risk of bias and applicability assessment

The 'Prediction model study Risk Of Bias Assessment Tool' (PROBAST) was used to assess the risk of bias and concerns about the applicability of the included studies [25]. The four domains of this tool, which are participants, predictors, outcome, and analysis were addressed. The overall judgement for the risk of bias evaluation and concern of applicability of the prediction models in PROBAST is 'low,' 'high,' or 'unclear.' In cases when all domains were graded 'low' risk of bias, assessment of 'models developed without external validation' was downgraded to 'high' risk of bias even if all the four domains were of low risk of bias, unless the model's development was based on an exceptionally large sample size and included some form of internal validation. External validation was considered if the model was at least validated using a dataset from a later time point in the same data source (temporal external validation) or using a different dataset from inside or outside the source country (geographical or broad external validation, respectively) [20]. Results of risk of bias and concern of applicability assessments were presented in a color-coded graph.

## Results

Our search strategy yielded 23,045 publications. After duplicate removal, 19,280 publication abstracts were screened, and 167 publications were eligible for full text screening. A total of 106 publications met our inclusion criteria (Fig 1). A list of the excluded studies with the justification of exclusion is presented (S1 Table). The results of the data extracted in this review are presented in the following subsections: geographical and chronological characteristics of the included studies, studies' type and design, and the ML models addressed, and (frequency of) health conditions investigated.

### Geographical and chronological characteristics

The earliest included study was published in 2002 [26], with the most publications occurring over the past four years. Most (77.3%, n = 82/106) of the publications were published between 2018–2021 (Fig 2). The United States of America (US) and the United Kingdom (UK) were reported in 57.1% of the included publications. While the 106 included publications reported countries 126 times, the US was reported 41 times and the UK 31 times. Usage of exclusive real-world PHC data for modelling was reported in 77.7% (n = 115 of 148 counts of data sources) across the studies. The remaining 22.3% of the PHC data sources were linked to different data sources, such as health insurance claims, cancer registries, secondary or tertiary health care, or administrative data. In the US, data were obtained mainly from PHC centers. In contrast, the most common source of the UK data were the Clinical Practice Research Datalink (CPRD), which is the largest patients' data registry in the UK [27]. The overall time span of health data across the studies ranged from 1982 [28] to 2020 [29]. The individual time span of the included studies varied between 2 months to 28 years. Sample sizes across the included studies ranged from 75 [30] to around 4 million [31] participants. The total number of the

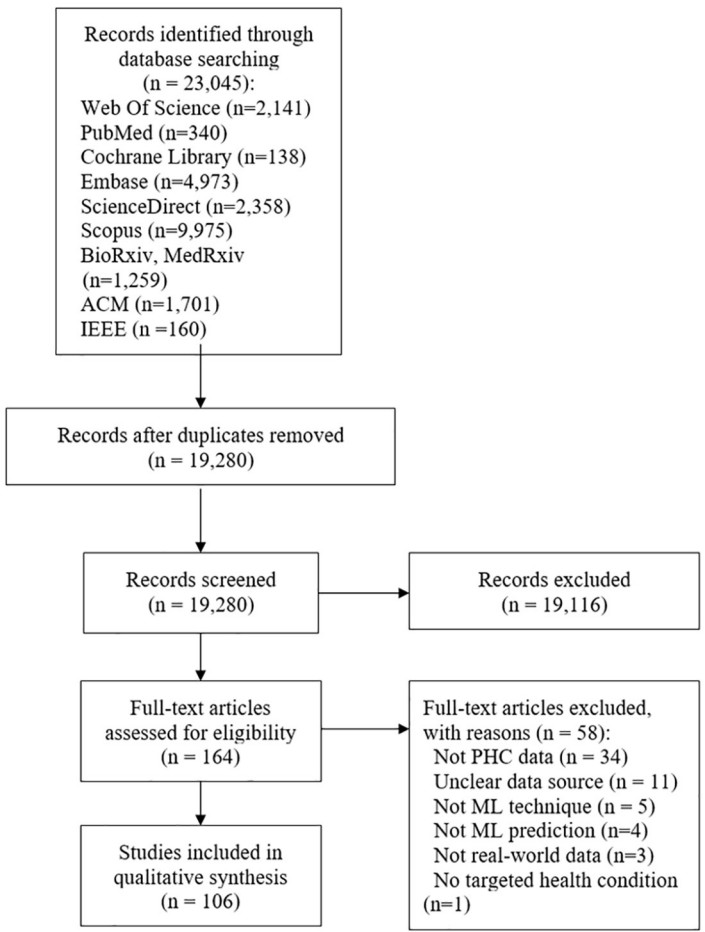

**Fig 1. Prisma flow diagram.**

populations within all the included studies was of 23.2 million. After correcting the potential overlaps, the total number of unique populations was reduced to be 22.7 million.

## Studies type and design, and ML models

The main type of the included studies was prediction models development without external validations (76.5%, n = 81 of 106). Of the remaining 25 studies, 13 studies (12.2%) developed and externally validated the models, and 12 studies (10.3%) externally validated previously existing models. Temporal validation [30, 32–36], geographical validation [37, 38], and using different population sample validations [39–44] were reported but none of these studies reported updating the assessed model.

All of the included studies were observational in design. Apart from 8 prospective studies, 92.4% (n = 98 of 106) of the studies were retrospective in design. Of the retrospective studies, 63 were retrospective cohorts. The other reported study designs were case control (n = 29), nested case control (n = 3), and cross sectional (n = 3). The purpose of the models reported was diagnostic in 77.3% (n = 82 of 106) of the studies, either incident (n = 62 of 82) or prevalent (n = 20 of 82). The remaining 23.5% (n = 25 of 106), including one study with two purposes of the models [45]) predicted prognosis of health conditions, such as remission, improvement, complications, hospitalization, or mortality. Despite all studies included used

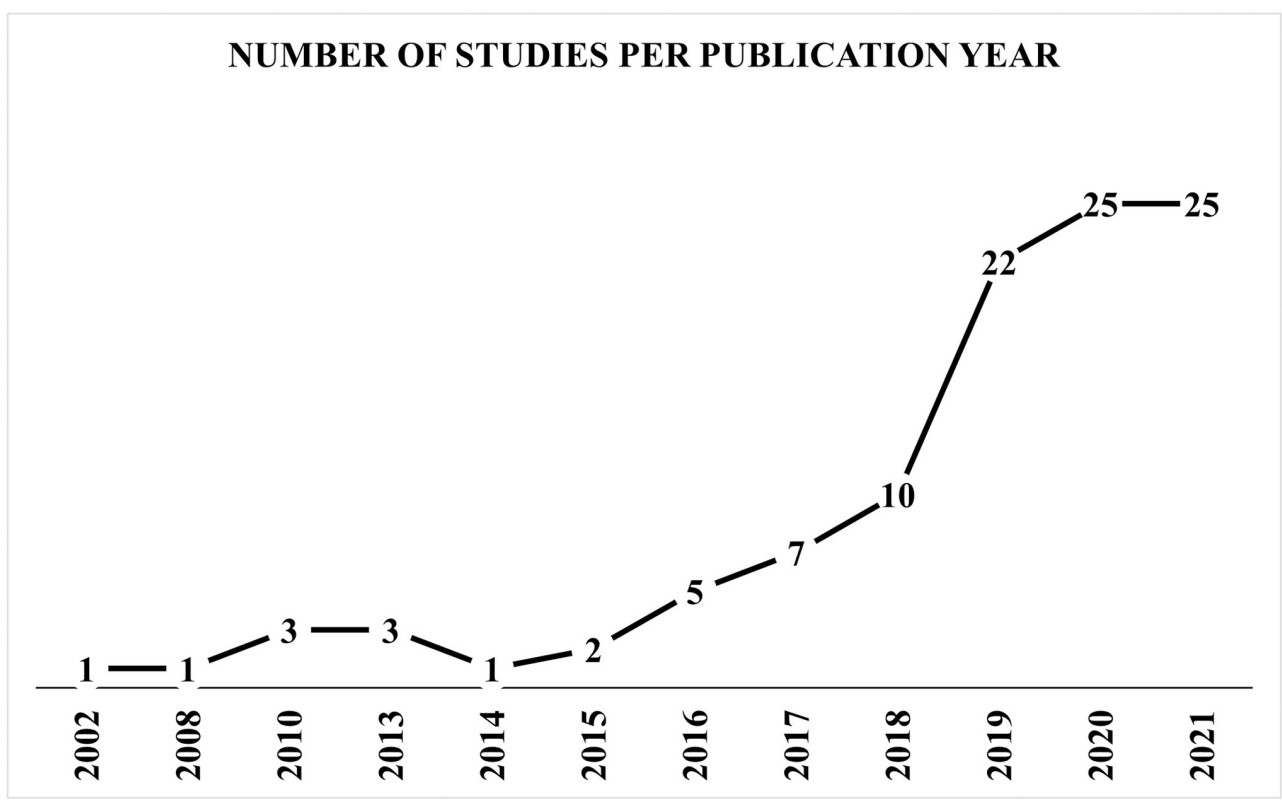

**Fig 2. Number of studies for publication years.**

real-world patients' data to develop and/or validate the ML models, four studies reported applying the models develop in real-world primary health care settings [46–48].

Within the 106 included publications, 207 models were developed and/or validated. The most frequently used type of ML was supervised learning 83.1% (n = 172 of 207 models across the included studies). These supervised ML models were identified as follows: random forest (n = 58), SVM (n = 30), boosting models such as extreme, light, and adaptive boosting (n = 28), decision tree (n = 25), and others such as naïve bias, k-nearest neighbors, and LASSO (n = 31). Deep learning techniques, such as neural networks, were reported 35 times (16.9%, of 207models), either exclusively or in comparison to other supervised ML models. Supplementary table (S2 Table) presents advantages and disadvantages of these models in addition to further descriptive results of our included studies. The most frequently reported evaluation approach of models' performance was the area under the receiver operating characteristic curve (AUROC), which was reported as "good" to "moderate" models performance in 62 studies. One study reported the performance measures using decision analysis curve [49]. Other evaluation approaches were reported across the included studies, such as calculating sensitivity, specificity, predictive values, and accuracy.

The data used to develop the models were called predictors, features, or variables across the included studies. These data were mostly textual. Demographic characteristics and clinical picture of the health conditions were the most frequently found data. Medications, comorbidities, and blood tests performed within primary care unit were reported. Data, such as blood test results and imaging results performed within secondary and tertiary health care were additionally reported in some of the individual studies. Referral documentation and clinical notes

taken by health care personnel were also reported. Five studies used the natural language processing (NLP) technique to handle free text clinical notes [40, 45, 50–52].

Tables 2–4 present an overview of the included studies characteristics based on the type of the study. They are grouped according to the ICD-10 classification and ordered alphabetically within each classification. A quantitative panel summary of all the included studies is also provided (S1 Panel).

### Health conditions

Out of the 22 classifications of the ICD-10, 11 classifications were addressed in the included studies. Frequently reported classifications were the endocrine, nutritional, and metabolic diseases classification (ICD-10: Class E00-E90) (n = 27 studies of 106, 25.5%), circulatory system diseases (ICD-10: Class I00-I99) (n = 23, 21.7%), and the mental and behavioral disorders classification (ICD-10: Class F00-F99) (n = 21, 19.9%). Diseases of the respiratory system classifications (ICD-10: Class J00-J99) and neoplasms (ICD-10: Class C00-C97) were addressed in (n = 10, 9.4% and n = 8, 7.5% respectively). 16% (n = 17) of the included studies investigated other health conditions (ICD 10: Classes G00-G99, K00-K93, M00-M99, N00-N99, O00-O99, and X60-X84).

**Endocrine, nutritional, and metabolic diseases (E00-E90).** In 27 studies addressing this classification [31, 34, 39, 46, 49, 50, 52, 72–86, 119, 127–130], populations involved were from 12 countries, mainly the US (41.9%). The studies were published since 2008 with the highest number of studies in 2019 (38.7%). 81% of the included studies reported the development and/or training of the proposed models using exclusive primary health care data of a total number of 4.2 million participants. Data were extracted from different sources covering a time span of six months to 23 years. Four health conditions were identified, namely diabetes mellitus (E10, E11) with/without complications (n = 21), familial hypercholesterolemia (E78) (n = 3), childhood obesity (E66) (n = 2), and primary aldosteronism (E26) (n = 1). Incident diagnostic prediction was the most commonly reported outcome (42%). Prevalent diagnostic and prognostic prediction were 32% and 26% respectively. Diabetic retinopathy was the most common complication (n = 5 of 21 related diabetes mellitus studies) reported. Diabetic foot was investigated in one study [50]. Two studies investigated prognostic predictive modelling of the short- and long-term levels of HbA1c after insulin treatment [49, 83].

**Mental and behavioral disorder (F00–F99).** In 21 studies addressing six health conditions [28, 30, 35, 92–107, 120, 121, 133], the populations were from eight countries, mainly the US and the UK (n = 13). These 21 studies were published since 2013 with the highest number published in 2020 (44.4%). Data were collected from different data sources with time span of data from one year to 28 years. Alzheimer's disease (F00) was addressed in 12 studies for mostly incident or prevalent diagnosis, apart from three studies. Depression (F32) was tackled in three studies, one of which predicted depression prognosis within two years [92]. Psychosis (F29) [35] and anxiety (F41) in cancer survivors seeking care in PHC [97] were addressed in one study each. Lastly, one study used PHC data to predict any mental disorder using different ML models [104].

**Circulatory and respiratory health conditions (I00-I99 and J00-J99).** In 33 studies, populations involved were from 11 countries, mainly the US and the UK. The included studies were published since 2010 with the highest number in both groups published in 2020 (30.8%). Data were extracted from the different data sources over time span one month to 23 years. Six circulatory health conditions were identified in 23 studies [29, 36, 37, 40, 45, 53–70]. These conditions were hypertension (I10-I15) (n = 5), heart failure (I50) (n = 5), atrial fibrillation (I48) (n = 2), stroke (I64) (n = 2), atherosclerosis (I70) (n = 1), myocardial infarction (I21)

**Table 2. Overview of the included studies with the type of ML prediction models development without conducting external validation (n = 81).**

| Study | Study design | Models purpose | Country | Source of data | Sample size | Time span of data | Health condition |
|---|---|---|---|---|---|---|---|
| **Circulatory System Diseases** | | | | | | | |
| **Chen et al. 2019** [53] | Retro. nested case control | Incident diagnostic | United States | PHC | 34,502 | 05/2000-05/2013 | Heart failure |
| **Choi et al. 2017** [54] | Retro. case control | Incident diagnostic | United States | PHC | 32,787 | 05/2000-05/2013 | Heart failure |
| **Du et al. 2020** [55] | Retro. cohort | Prognostic | China | Linked data | 42,676 | 2010–2018 | Hypertension |
| **Farran et al. 2013** [56] | Retro. cohort | Incident diagnostic | Kuwait | Linked data | 270,172 | 12 years | Any cardiovascular disease |
| **Hill et al. 2019** [57] | Retro. cohort | Incident diagnostic | United Kingdom | PHC | 2,994,837 | 01/2006-12/2016 | Atrial fibrillation |
| **Karapetyan et al. 2021** [29] | Retro. cohort | Prognostic | Germany | PHC | 46,071 | 02-2020-09/2020 | Any cardiovascular disease |
| **Lafreniere et al. 2017** [58] | Retro. cohort | Incident diagnostic | Canada | PHC | 379,027 | Not reported | Hypertension |
| **Li et al. 2020** [59] | Retro. cohort | Incident diagnostic | United Kingdom | Linked data | 3,661,932 | 01/1998-12/2018 | Any cardiovascular disease |
| **Lip et al. 2021** [60] | Retro. cohort | Prevalent diagnostic | Australia | Linked data | 926 | Not reported | Hypertension |
| **Lorenzoni et al. 2019** [61] | Pros. cohort | Prognostic | Italy | Linked data | 380 | 2011–2015 | Heart failure |
| **Ng et al. 2016** [62] | Retro. nested case control | Incident diagnostic | United States | PHC | 152,095 | 2003–2010 | Heart failure |
| **Nikolaou et al. 2021** [63] | Retro. cohort | Prognostic | United Kingdom | PHC | 6,883 | 2015–2019 | Any cardiovascular disease |
| **Sarraju et al. 2021** [64] | Retro. cohort | Incident diagnostic | United States | PHC | 32,192 | 01/2009-12/2018 | Any cardiovascular disease |
| **Selskyy et al. 2018** [65] | Retro. case control | Prognostic | Ukraine | PHC | 63 | 2011–2012 | Hypertension |
| **Shah et al. 2019** [45] | Retro. cohort | Prognostic/Incident diagnostic | United Kingdom | Linked data | 2,000 | Not reported | Myocardial infarction |
| **Solanki et al. 2020** [66] | Retro. cohort | Prevalent diagnostic | United States | PHC | 495 | 2007–2017 | Hypertension |
| **Solares et al. 2019** [67] | Retro. cohort | Incident diagnostic | United Kingdom | PHC | 80,964 | entry– 01/2014 | Any cardiovascular disease |
| **Ward et al. 2020** [68] | Retro. cohort | Incident diagnostic | United States | PHC | 262,923 | 01/2009-12/2018 | Atherosclerosis |
| **Weng et al. 2017** [69] | Retro. cohort | Incident diagnostic | United Kingdom | PHC | 378,256 | 01/2005-01/2015 | Any cardiovascular disease |
| **Wu et al. 2010** [70] | Retro. case control | Incident diagnostic | United States | PHC | 44,895 | 01/2003-12/2006 | Heart failure |
| **Zhao et al. 2020** [40] | Retro. cohort | Incident diagnostic | United States | Linked data | 4,914 | Not reported | Stroke |
| **Digestive System Diseases** | | | | | | | |
| **Sáenz Bajo et al. 2002** [26] | Retro. cohort | Prevalent diagnostic | Spain | PHC | 81 | 01/1999-06/1999 | Gastroesophageal reflux |
| **Waljee et al. 2018** [71] | Retro. cohort | Prognostic | United States | Linked data | 20,368 | 2002–2009 | Inflammatory bowel disorders |
| **Endocrine, Metabolic, and Nutritional Diseases** | | | | | | | |
| **Akyea et al. 2020** [31] | Retro. cohort | Incident diagnostic | United Kingdom | PHC | 4,027,775 | 01/1999-06/2019 | Familial hypercholesterolemia |
| **Á-Guisasola et al. 2010** [72] | Pros. cohort | Incident diagnostic | Spain | PHC | 2,662 | Not reported | Diabetes mellitus |
| **Crutzen et al. 2021** [73] | Retro. cohort | Incident diagnostic | The Netherlands | PHC | 138,767 | 01/2007-01/2014 | Diabetes mellitus |
| **Ding et al. 2019** [74] | Retro. case control | Prevalent diagnostic | United States | PHC | 97,584 | 1997–2017 | Primary Aldosteronism |
| **Dugan et al. 2015** [75] | Retro. cohort | Prognostic | United States | PHC | 7,519 | Over 9 years | Obesity |
| **Farran et al. 2019** [76] | Retro. cohort | Prognostic | Kuwait | PHC | 1,837 | Over 9 years | Diabetes mellitus |

*(Continued)*

**Table 2.** (*Continued*)

| Study | Study design | Models purpose | Country | Source of data | Sample size | Time span of data | Health condition |
|---|---|---|---|---|---|---|---|
| **Hammond *et al*. 2019** [77] | Retro. cohort | Prognostic | United States | PHC | 3,449 | 01/2008-08/2016 | Obesity |
| **Kopitar *et al*. 2020** [78] | Retro. case control | Incident diagnostic | Slovenia | PHC | 27,050 | 12/2014-09/2017 | Diabetes mellitus |
| **Lethebe *et al*. 2019** [79] | Retro. cohort | Prevalent diagnostic | Canada | PHC | 1,309 | 2008–2016 | Diabetes mellitus |
| **Looker *et al*. 2015** [80] | Retro. case control | Prognostic | United Kingdom | PHC | 309 | 12/1998-05/2009 | Diabetic nephropathy |
| **Metsker *et al*. 2020** [81] | Retro. cohort | Incident diagnostic | Russia | NR | 54,252 | 07/2009-08/2017 | Diabetic polyneuropathy |
| **Metzker *et al*. 2020** [82] | Retro. cohort | Incident diagnostic | Russia | NR | 58,462 | Not reported | Diabetic polyneuropathy |
| **Nagaraj *et al*. 2019** [83] | Retro. cohort | Prognostic | The Netherlands | PHC | 11,887 | 01/2007-12/2013 | Diabetes mellitus |
| **Pakhomov *et al*. 2008** [50] | Retro. cohort | Prevalent diagnostic | United States | PHC | 145 | 07/2004-09/2004 | Diabetic foot |
| **Rumora *et al*. 2021** [84] | Cross sectional | Incident diagnostic | Denmark | PHC | 97 | 10/2015-06/2016 | Diabetic polyneuropathy |
| **Tseng *et al*. 2021** [52] | Cross sectional | Incident diagnostic | United States | PHC | NR | 07/2016-12/2018 | Diabetes mellitus |
| **Wang *et al*. 2021** [85] | Retro. cohort | Incident diagnostic | China | PHC | 1,139 | 2017–2019 | Gestational diabetes |
| **Williamson *et al*. 2020** [86] | Pros. cohort | Incident diagnostic | United States | Linked data | 866 | Not reported | Familial hypercholesterolemia |
| **External Cause of Mortality** | | | | | | | |
| **DelPozo-Banos *et al*. 2018** [87] | Retro. case control | Incident diagnostic | United Kingdom | Linked data | 54,684 | 2001–2015 | Suicidality |
| **Penfold *et al*. 2021** [88] | Retro. cohort | Incident diagnostic | United States | Linked data | 256,823 | Not reported | Suicidality |
| **van Mens *et al*. 2020** [89] | Retro. case control | Incident diagnostic | The Netherlands | PHC | 207,882 | 2017 | Suicidality |
| **Genitourinary System Diseases** | | | | | | | |
| **Shih *et al*. 2020** [90] | Retro. cohort | Incident diagnostic | Taiwan | Linked data | 19,270 | 01/2015-12/2019 | Chronic kidney disease |
| **Zhao *et al*. 2019** [91] | Retro. cohort | Incident diagnostic | United States | PHC | 61,740 | 2009–2017 | Chronic kidney disease |
| **Mental and Behavioral Diseases** | | | | | | | |
| **Dinga *et al*. 2018** [92] | Pros. cohort | Prognostic | The Netherlands | Linked data | 804 | Not reported | Depression |
| **Ford *et al*. 2019** [93] | Retro. case control | Incident diagnostic | United Kingdom | PHC | 93,120 | 2000–2012 | Alzheimer's disease |
| **Ford *et al*. 2020** [94] | Retro. case control | Incident diagnostic | United Kingdom | PHC | 95,202 | 2000–2012 | Alzheimer's disease |
| **Ford *et al*. 2021** [95] | Retro. case control | Prevalent diagnostic | United Kingdom | PHC | 93,426 | 2000–2012 | Alzheimer's disease |
| **Fouladvand *et al*. 2019** [96] | Retro. cohort | Prognostic | United States | PHC | 3,265 | Not reported | Alzheimer's disease |
| **Haun *et al*. 2021** [97] | Cross sectional | Incident diagnostic | Germany | PHC | 496 | Not reported | Anxiety |
| **Jammeh *et al*. 2018** [98] | Retro. case control | Incident diagnostic | United Kingdom | PHC | 3,063 | 06/2010-06/2012 | Alzheimer's disease |
| **Jin *et al*. 2019** [99] | Retro. cohort | Incident diagnostic | United States | PHC | 923 | 2010–2013 | Depression |
| **Kaczmarek *et al*. 2019** [100] | Retro. case control | Prevalent diagnostic | Canada | PHC | 890 | Not reported | Post-traumatic stress disorder |
| **Ljubic *et al*. 2020** [101] | Retro. cohort | Incident diagnostic | United States | PHC | 2,324 | Not reported | Alzheimer's disease |
| **Mallo *et al*. 2020** [102] | Retro. case control | Prognostic | Spain | PHC | 128 | 2008 | Alzheimer's disease |
| **Mar *et al*. 2020** [103] | Retro. case control | Prevalent diagnostic | Spain | Linked data | 4,003 | Not reported | Alzheimer's disease |

(*Continued*)

**Table 2.** (Continued)

| Study | Study design | Models purpose | Country | Source of data | Sample size | Time span of data | Health condition |
|---|---|---|---|---|---|---|---|
| **Półchłopek et al. 2020** [104] | Retro. cohort | Incident diagnostic | The Netherlands | PHC | 92,621 | 2007-12/2016 | Any mental disorder |
| **Shen et al. 2020** [105] | Retro. cohort | Incident diagnostic | China | PHC | 2,299 | 2008–2018 | Alzheimer's disease |
| **Suárez-Araujo et al. 2021** [106] | Retro. case control | Prevalent diagnostic | United States | PHC | 330 | Not reported | Alzheimer's disease |
| **Tsang et al. 2021** [28] | Retro. cohort | Prognostic | United Kingdom | PHC | 59,298 | 1982–2015 | Alzheimer's disease |
| **Zafari et al. 2021** [107] | Retro. cohort | Incident diagnostic | Canada | PHC | 154,118 | 01/1995-12/2017 | Post-traumatic stress disorder |
| **Musculoskeletal and Connective Tissue Diseases** | | | | | | | |
| **Emir et al. 2014** [108] | Retro. cohort | Incident diagnostic | United States | PHC | 587,961 | 2011–2012 | Fibromyalgia |
| **Jarvik et al. 2018** [109] | Pros. cohort | Prognostic | United States | PHC | 3,971 | 03/2011-03/2013 | Back pain |
| **Kennedy et al. 2021** [110] | Retro. case control | Incident diagnostic | United Kingdom | Linked data | 23,528 | Over 6 years | Ankylosing spondylitis |
| **Neoplasms** | | | | | | | |
| **Kop et al. 2016** [111] | Retro. cohort | Incident diagnostic | The Netherlands | PHC | 260,000 | Not reported | Colorectal cancer |
| **Malhotra et al. 2021** [112] | Retro. case control | Incident diagnostic | United Kingdom | PHC | 5,695 | 01/2005-06/2009 | Pancreatic cancer |
| **Ristanoski et al. 2021** [113] | Retro. case control | Incident diagnostic | Australia | PHC | 683 | 2016–2017 | Lung cancer |
| **Nervous System Diseases** | | | | | | | |
| **Cox et al. 2016** [114] | Retro. case control | Prevalent diagnostic | United Kingdom | PHC | 3,960 | 01/2007-12/2011 | Post stroke spasticity |
| **Hrabok et al. 2021** [115] | Retro. cohort | Prognostic | United Kingdom | PHC | 10,499 | 01/2000-05/2012 | Epilepsy |
| **Kwasny et al. 2021** [116] | Retro. case control | Incident diagnostic | Germany | PHC | 3,274 | 01/2010-12/2017 | Progressive supranuclear palsy |
| **Respiratory System Diseases** | | | | | | | |
| **Afzal et al. 2013** [117] | Retro. cohort | Prevalent diagnostic | The Netherlands | PHC | 5,032 | 01/2000-01/2012 | Asthma |
| **Doyle et al. 2020** [118] | Retro. case control | Incident diagnostic | United Kingdom | PHC | 112,784 | 09/2003-09/2017 | Non-tuberculous mycobacterial lung |
| **Kaplan et al. 2020** [32] | Retro. cohort | Prevalent diagnostic | United States | Linked data | 411,563 | Not reported | Asthma/obstructive pulmonary disease |
| **Lisspers et al. 2021** [41] | Retro. cohort | Prognostic | Sweden | Linked data | 29,396 | 01/2000-12/2013 | Asthma |
| **Marin-Gomez et al. 2021** [42] | Retro. cohort | Incident diagnostic | Spain | PHC | 7,314 | 03/04/2020 | COVID-19 |
| **Ställberg et al. 2021** [33] | Retro. cohort | Prognostic | Sweden | Linked data | 7,823 | 01/2000-12/2013 | Chronic obstructive pulmonary disease |
| **Stephens et al. 2020** [51] | Retro. case control | Incident diagnostic | United States | PHC | 7,278 | 2009–2019 | Influenza |
| **Trtica-Majnaric et al. 2010** [43] | Retro. cohort | Prognostic | Croatia | PHC | 90 | 2003–2004 | Influenza |
| **Zafari et al. 2022** [44] | Retro. cohort | Incident diagnostic | Canada | PHC | 4,134 | Not reported | Chronic obstructive pulmonary disease |

**Table 3. Overview of the included studies with the type of ML prediction models development with conduction of external validation (n = 13).**

| Study | Study design | Models purpose | Country | Source of data | Sample size | Time span of data | Health condition |
|---|---|---|---|---|---|---|---|
| **Endocrine, Metabolic, and Nutritional Diseases** | | | | | | | |
| **Hertroijs *et al.* 2018** [49][a] | Retro. cohort | Prognostic | The Netherlands | PHC | 10,528 | 01/2006-12/2014 | Diabetes mellitus |
| | | | The Netherlands | PHC | 3,337 | 01/2009-12/2013 | |
| **Myers *et al.* 2019** [39] [a] | Retro. case control | Incident diagnostic | United States | PHC | 33,086 | 09/2013-08/2016 | Familial hypercholesterolemia |
| | | | United States | Linked data | 7,805 | | |
| | | | United States | Linked data | 35,090 | | |
| | | | United States | Linked data | 8,094 | | |
| **Perveen *et al.* 2019** [34] [a] | Retro. cohort | Prognostic | Canada | PHC | 911 | 08/2003-06/2015 | Diabetes mellitus |
| | | | Canada | PHC | 1,970 | | |
| **Weisman *et al.* 2020** [119] [a] | Retro. cohort | Prevalent diagnostic | Canada | PHC | 5,402 | 2010–2017 | Diabetes mellitus |
| | | | Canada | Linked data | 29,371 | | |
| **Mental and Behavioral Diseases** | | | | | | | |
| **Amit *et al.* 2021** [120] [a] | Retro. cohort | Prevalent diagnostic | United Kingdom | PHC | 24,612 | 2000–2010 | Post-partum depression |
| | | | United Kingdom | PHC | 9,193 | 2010–2017 | |
| | | | United Kingdom | PHC | 34,525 | 2000–2017 | |
| **Levy *et al.* 2018** [30] [a] | Retro. cohort | Incident diagnostic | United States | PHC | 49 | Over 9 months | Alzheimer's disease |
| | | | United States | Linked data | 26 | Not reported | |
| **Perlis 2013** [121] [a] | Retro. cohort | Prognostic | United States | PHC | 2,094 | 1999–2006 | Depression |
| | | | United States | PHC | 461 | | |
| **Raket *et al.* 2020** [35] [a] | Retro. case control | Incident diagnostic | United States | PHC | 145,720 | 1990–2018 | Psychosis |
| | | | United States | PHC | 4,770 | | |
| **Musculoskeletal and Connective Tissue Diseases** | | | | | | | |
| **Fernandez-Gutierrez *et al.* 2021** [122] [a] | Retro. cohort | Incident diagnostic | United Kingdom | Linked data | 19,314 | 2002–2012 | Rheumatoid arthritis & Ankylosing spondylitis |
| | | | United Kingdom | Linked data | 1,868 | | |
| **Jorge *et al.* 2019** [123] [a] | Retro. cohort | Incident diagnostic | United States | Linked data | 400 | Not reported | Systematic lupus erythematous |
| | | | United States | Linked data | 173 | Not reported | |
| **Zhou *et al.* 2017** [124] [a] | Retro. cohort | Incident diagnostic | United Kingdom | Linked data | Not reported | 10/2013-07/2014 | Rheumatoid arthritis |
| | | | United Kingdom | Linked data | 475,580 | 03/2009-10/2012 | |
| **Neoplasms** | | | | | | | |
| **Kinar *et al.* 2016** [125] [a] | Retro. cohort | Incident diagnostic | Israel | PHC | 606,403 | 01/2003-07/2011 | Colorectal cancer |
| | | | United Kingdom | PHC | 30,674 | 01/2003-05/2012 | |
| **Pregnancy, Childbirth, Puerperium** | | | | | | | |
| **Sufriyana *et al.* 2020** [126] [a] | Retro. nested case control | Incident diagnostic | Indonesia | Linked data | 20,975 | 2015–2016 | Preeclampsia |
| | | | Indonesia | Linked data | 1,322 | Not reported | |
| | | | Indonesia | Linked data | 904 | Not reported | |

[a]Each row per study represents a different dataset that was used to develop and/or validate the prediction models.

**Table 4. Overview of the included studies with the type of reporting external validation of previously developed ML prediction models (n = 12).**

| Study | Study design | Models purpose | Country | Source of data | Sample size | Time span of data | Health condition |
|---|---|---|---|---|---|---|---|
| **Circulatory System Diseases** | | | | | | | |
| **Kostev et al. 2021** [37] | Retro. cohort | Incident diagnostic | Germany | PHC | 11,466 | 01/2010-12/2018 | Stroke |
| **Sekelj et al. 2020** [36] | Retro. cohort | Incident diagnostic | United Kingdom | PHC | 604,135 | 01/2001-12/2016 | Atrial fibrillation |
| **Endocrine, Metabolic, and Nutritional Diseases** | | | | | | | |
| **Abramoff et al. 2019** [127] | Pros. cohort | Prevalent diagnostic | United States | PHC | 819 | 01/2017-07/2017 | Diabetic retinopathy |
| **Bhaskaranand et al. 2019** [128] | Retro. cohort | Prevalent diagnostic | United States | PHC | 1,017,001 | 01/2014-09/2015 | Diabetic retinopathy |
| **González-Gonzalo et al. 2019** [129] [a] | Retro. case control | Prevalent diagnostic | Spain | PHC | 288 | 08/2011-10/2016 | Diabetic retinopathy |
| | | | Sweden | PHC | | | |
| | | | Denmark | PHC | | | |
| | | | United States | Linked data | 4,613 | Over 2014 | |
| | | | United Kingdom | Linked data | | | |
| **Kanagasingam et al. 2018** [130] | Pros. cohort | Incident diagnostic | Australia | PHC | 193 | 12.2016-05/2017 | Diabetic retinopathy |
| **Verbraak et al. 2019** [46] | Retro. cohort | Prevalent diagnostic | The Netherlands | PHC | 1,425 | 2015 | Diabetic retinopathy |
| **Neoplasms** | | | | | | | |
| **Birks et al. 2017** [38] | Retro. case control | Incident diagnostic | United Kingdom | PHC | 2,550,119 | 01/2000-04/2015 | Colorectal cancer |
| **Hoogendoorn et al. 2016** [131] | Retro. case control | Prevalent diagnostic | The Netherlands | PHC | 90,000 | 07/2006-12/2011 | Colorectal cancer |
| **Hornbrook et al. 2017** [47] | Retro. case control | Incident diagnostic | United States | Linked data | 17,095 | 1998–2013 | Colorectal cancer |
| **Kinar et al. 2017** [48] | Pros. cohort | Incident diagnostic | Israel | Linked data | 112,584 | 07/2007-12/2007 | Colorectal cancer |
| **Respiratory System Diseases** | | | | | | | |
| **Morales et al. 2018** [132] | Retro. cohort | Prognostic | United Kingdom | PHC | 2,044,733 | 01/2000-04/2014 | Chronic obstructive pulmonary disease |

[a]Each row per study represents a different dataset that was used to develop and/or validate the prediction models.

(n = 1), and any cardiovascular event or disease (n = 7). Five respiratory health conditions were investigated in 10 studies [32, 33, 41–44, 51, 117, 118, 132, 134, 135]. Four studies predicted mortality and hospitalization risks on top of chronic obstructive pulmonary disease (COPD) (J40). Two studies investigated prevalent diagnosis of Asthma (J45) and its exacerbation risk. Influenza was predicated in two studies [117, 124] for incident cases and prognosis. COVID-19 (U07) incident cases were predicted within routine PHC visits in one study [42].

**Other health conditions.** Eight studies colorectal cancer (CRC) (C18) (n = 6), lung cancer (C34) (n = 1), and pancreatic cancer (C25) (n = 1). Four studies addressed the same incidence prediction model known as ColonFlag (previously MeScore) to identify CRC cases [38, 47, 48, 125]. Each study predicted incident cases within different time windows before diagnosis; from three months to two years. Three health conditions affecting the nervous system were addressed [114–116], which were post stroke spasticity, epilepsy specifically mortality four years before and after its diagnosis (G40) [115], and a rare neurodegenerative disease progressive supra-nuclear palsy (G23) [116]. A few studies investigated musculoskeletal and

connective tissue disorders as well as gastrointestinal and kidney diseases [122–124, 108–110]. The musculoskeletal and connective tissue condition were back pain (M54) prognosis within PHC settings [109], ankylosing spondylitis (M45) [110]. The gastrointestinal and kidney diseases were examined in four studies, namely inflammatory bowel diseases (K50-K52), including Crohn's disease and ulcerative colitis [26, 71], peptic ulcers (K27)/gastroesophageal reflux (K21), and chronic kidney disease (N18) [90, 133]. Three studies tackled suicidality (X60-X84) [87–89]. Lastly, one study addressed preeclampsia (O14) [126].

## Quality assessment

Quality was assessed using the PROBAST tool and 90.5% (n = 96 of 106) of the included studies were of high and unclear risk of bias (Fig 3). Analysis domain was the main source of bias, because of underreporting. It was found that only a few studies (n = 11) were reported in accordance with transparent reporting of a multivariable prediction model for individual prognosis or diagnosis (TRIPOD) guidelines [136]. Nevertheless, studies of low risk of bias were downgraded to be of high risk of bias due to the of lack of external validation of the proposed models (n = 20). The second concern assessed using this tool was the concern of applicability, which was estimated as low to moderate concern (66%). The dependence of the predictive models on not-routine PHC data as a concern of models' applicability within PHC settings was raised in 34% of the studies.

Most of the included studies (n = 98 of 106, 92.5%) were published as peer-reviewed articles in biomedical (e.g., PLOS ONE, n = 8) and technical journals (e.g., IEEE, n = 3). Eight studies were preprint and abstracts. National research institutes and universities were the most frequently reported funding support. Most of the studies reported that the funders were not involved in the published work.

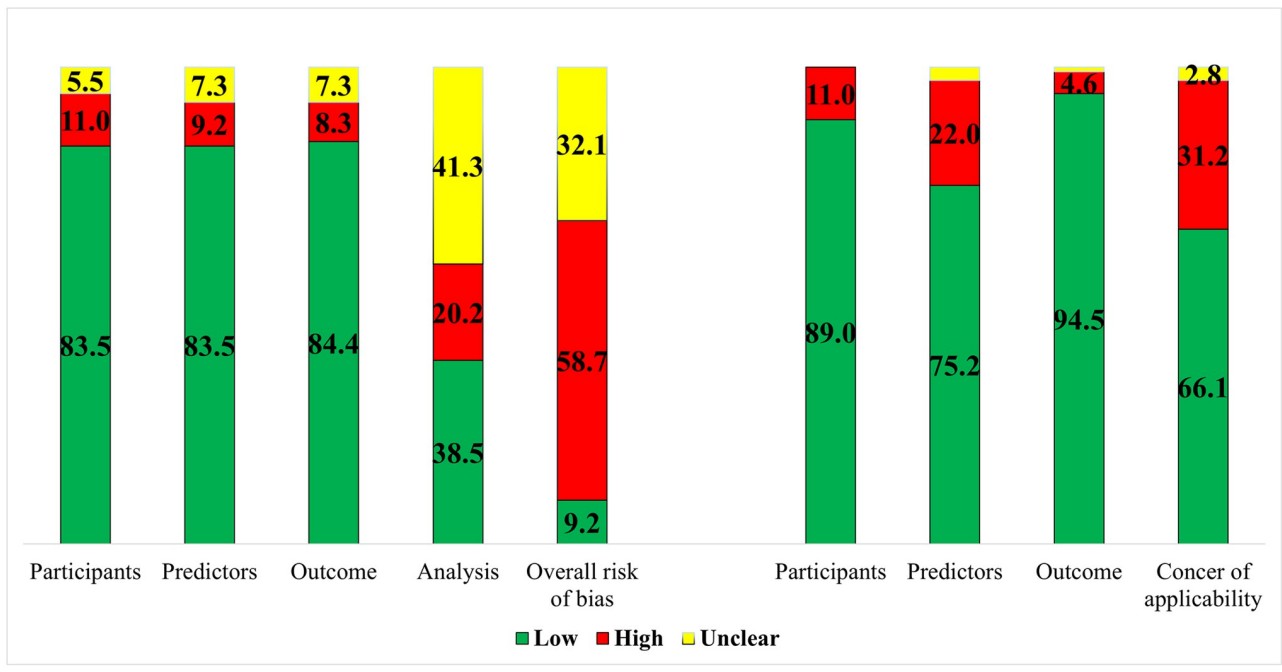

**Fig 3. Percentage presentation of the results of (PROBAST) tool.** The tool has two components. Component 1. Risk of bias (4 domains: Participants, predictors, outcome, and analysis). Component 2. Concern of applicability (3 domains: Participants, predictors, and outcome).

## Discussion

ML prediction models could have an immense potential to augment health practice and clinical decision making in various health sectors. Our systematic review provides an outline of the health conditions investigated with ML prediction models using PHC data.

### Summary of findings

In 106 observational studies, we identified 42 health conditions targeted by 207 ML prediction models, of which 42.5% were random forest and SVM. The included models used PHC data documented over the past 40 years for a total of 22.7 million patients. Half of the studies were conducted in the US and the UK. While the majority of the included studies (77.3%) focused on diagnosis prediction, a significant portion also addressed predictive aspects related to complications, hospitalization, and mortality. The most frequently targeted health conditions included Alzheimer's disease, diabetes mellitus, heart failure, colorectal cancer, and chronic obstructive pulmonary diseases, while other conditions such as asthma, childhood obesity, and dyspepsia received comparatively less attention. A considerable portion of the models (76.4% of the included studies) were trained and internally validated without evaluating their generalizability.

### Results in perspective

Detection and management of health conditions, particularly those that are preventable and controllable like diabetes mellitus, stand for the fundamental role of PHC [3]. Advances in such technologies might enhance health care and quality of life. Noticeably, they have gained more attention in many countries [11]. Our findings of common and rare health conditions targeted by ML prediction models in PHC indicates increase of research interest. However, clinical implication of such models is still limited to the theoretical good performance. Furthermore, the unequal distribution of publications across countries could be related to the low publication rate or lack of proper health data documentation systems in lower income countries, which impose further limitation to validate and implement such models.

The coding system used in health records does not universally follow the same criteria for all diseases, posing challenges for the consistency of models' performance [137]. Moreover, the lack of globally standardized definitions and terminology of diseases and the wide variability of the services provided across different health systems further limit the effectiveness of the models [137]. For example, uncoded free-text clinical notes as well as using 'race' and 'ethnicity' or 'suicide' and 'suicide attempts' to be documented as a single input can affect the predictive power of the models [138]. Other drawbacks reported include underrepresentation of healthy persons, retrospective temporal dimension of predictors, and the absence of confirmatory diagnostic services in PHC pose significant limitations [139, 140].

Technical biases can significantly influence the clinical utility of technologies. Models trained on historical data without adaptation to policy changes may reinforce outdated practices, leading to erroneous results [141]. Additionally, validating models using different populations data can create a mismatch between the data or environment on which the models was trained; this mismatch may impact the accuracy of the models' prediction [141]. Therefore, documenting characteristics of the health systems may highlight the discrepancies between the data used to train and validate the models. This may improve the validation and implementation processes of the models. Models that are known for their high prediction accuracy, such as random forest and SVM might support better health outcomes when developed using high quality health data [139]. Additionally, the variety of the ML prediction models characteristics provide opportunities to improve healthcare practice. Using large data documented as

electronic health records, random forest models and ensemble models such as boosting models have the ability to handle large datasets with numerous predictors variables [140]. Artificial neural network can also perform complex images processing that can boost the primary health care services [140]. Furthermore, SVM and decision tree models can provide nonlinear solutions, thus will support our understanding of complex and dynamic diseases for earlier health conditions prediction [142].

Nature of diseases append further challenges. The most challenging diseases for ML prediction are multifaceted long-term health conditions, such as DM, that are influenced by combination of genetic, environmental, and lifestyle factors. The complex health conditions further tangle the models, making it harder to identify accurate predictive patterns. Furthermore, the subjective nature of symptoms, especially symptoms related to mental health disorders, pose additive challenges toward ML models accuracy. Rare diseases, if documented, often suffer from limited data availability, leading to difficulty to train ML models effectively [143].

Health care professionals are fundamental to the process of implementing and integrating ML prediction models in their healthcare practice. Despite that, our review did not report outcomes related to healthcare professionals. Significant variability of opinions on the utilization of ML in PHC among primary health care providers hinder its acceptance. Furthermore, the black-box nature of ML prediction models precludes the clinical interpretability of models' outcomes. Additional workload and training are needed to implement such technology in the routine practice. Trust, data protection, and ethical and clinical responsibility legislation are further intractable issues that represent major obstacles toward ML prediction models implementation [5].

A considerable lack of usage of studies reporting guidelines across the included studies lead to deficient description of the populations' demographics and underreporting of the models' related statistical analysis, which lead to high risk of bias of majority of studies. These shortcomings negatively affect the reproducibility of the models [144]. Navarro and colleagues investigated this underreporting, and they claimed that the available reporting guidelines of modelling studies might be less apposite for ML models studies [145].

## Implication of results and recommendation for future contributions

This review provided a comprehensive outline of ML prediction models in PHC and raises important considerations for future research and implementation of this technology in PHC settings. Interdisciplinary collaboration among health care workers, developers of ML models, and creators of digital documentation systems is required. This is especially important given the increasing popularity of digitally connected health systems [5]. It is recommended to augment the participation of health professionals through the development process of the PHC predictive models to critically evaluate, assess, adopt, and challenge the validation of the models within practices. This collaboration may assist ML engineers to recognize unintended negative implications of their algorithms, such as accidentally fitting of confounders, and unintended discriminatory bias, among others, for better health outcomes [146]. Health care systems need to provide comprehensive population health data repositories as an enabler for medical analyses [137]. Well-designed and -documented repositories which provide representative health data for the healthy and diseased populations are needed [137, 139]. These high-quality data repositories might provide future modelling studies with data that match the studies' clinical research questions for more accurate prediction. Further ML prediction studies are needed to target more health conditions using PHC data. Despite the additional burden, it is beneficial also to continuously assess the potential significance of models, such as improved

health outcomes, reduced medical errors, increased professional effectiveness and productivity, and enhanced patients' quality of life [147]. It is recommended to follow reporting guidelines for producing valid and reproducible ML modelling studies. Developing robust frameworks to enable the adoption and integration of ML models in the routine practice is also essential for effective transition from conventional health care systems to digital health [148, 149]. Sophisticated technical infrastructure and strong academic and governmental support are essential for promoting and supporting long-term and broad-reaching PHC ML-based services [138, 150]. However, balanced arguments [151, 152] regarding the potential benefits and limitations of ML models support better health care without overestimating or hampering the use of such technology. It is also suggested to integrate the basic understanding of ML concepts and techniques in education programs for health science and medical students.

## Strengths and limitations of the review

Our review was conducted following a predesigned comprehensive protocol [21]. We identified the health conditions targeted within PHC settings and identified the gaps that need to be addressed. The main limitation of our review is the low quality of evidence of the primary evidence. It is also possible due to the wide array of descriptors that exist to describe ML, our search strategy could have missed some studies if they exclusively used terms outside of our search string [153]. Limiting the scope of our review to clinical health conditions might have excluded other conditions, such as domestic violence and drug abuse [3]. Guiding our work using ICD-10 might have led to the exclusion of some health conditions, such as frailty studies [154]. Lastly, we did not present the statistical analysis of the models' attributes or conduct a meta-analysis, because of the broad heterogeneity across studies. In the future, we plan to update our review–considering the noticeable rise of ML studies within PHC, while also modifying our methodology to reduce the identified limitations. It is also planned to use the specific ML guidelines TRIPOD-AI and PROBAST-AI when published to strengthen quality and reporting of our findings [155].

In conclusion, ML prediction models within PHC are gaining traction. Further studies examining the use of ML in real PHC settings are needed, especially those with prospective designs and more representative samples. Collaborating amongst multidisciplinary teams to tackle ML in PHC will increase the confidence in models and their implementations in clinical practice.

## Supporting information

**S1 Table. List of excluded studies with reasons (n = 58).**
(PDF)

**S2 Table. Characteristics of the included ML predictive models.**
(PDF)

**S1 File. Search strategy.**
(PDF)

**S2 File. Prisma checklist.**
(DOC)

**S1 Panel. Quantitative summary of the included studies' characteristics (n = 106).**
(PDF)

## Acknowledgments

Marcos André Gonçalves, PhD and Bruna Zanotto, MSc, provided their feedback on the project's primary draft. Luana Fiengo Tanaka, PhD, helped retrieved inaccessible studies.

## Author Contributions

**Conceptualization:** Hebatullah Abdulazeem.

**Data curation:** Hebatullah Abdulazeem, Sera Whitelaw.

**Formal analysis:** Hebatullah Abdulazeem.

**Methodology:** Hebatullah Abdulazeem.

**Project administration:** Hebatullah Abdulazeem.

**Supervision:** Gunther Schauberger, Stefanie J. Klug.

**Visualization:** Hebatullah Abdulazeem.

**Writing – original draft:** Hebatullah Abdulazeem.

**Writing – review & editing:** Hebatullah Abdulazeem, Sera Whitelaw, Gunther Schauberger, Stefanie J. Klug.

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
