## [Decision Letter · Decision Letter 0]

5 Apr 2023

PONE-D-22-23382A systematic review of clinical health conditions predicted by machine learning diagnostic and prognostic models trained or validated using real-world primary health care dataPLOS ONE

Dear Dr. Abdulazeem,

Thank you for submitting your manuscript to PLOS ONE. After careful consideration, we feel that it has merit but does not fully meet PLOS ONE’s publication criteria as it currently stands. Therefore, we invite you to submit a revised version of the manuscript that addresses the points raised during the review process.

We look forward to receiving your revised manuscript.

Kind regards,

Ágnes Vathy-Fogarassy, Ph.D.

Academic Editor

PLOS ONE

Journal Requirements:

Reviewers' comments:

Reviewer's Responses to Questions

Comments to the Author

1. Is the manuscript technically sound, and do the data support the conclusions?

Reviewer #1: Yes

Reviewer #2: Yes

2. Has the statistical analysis been performed appropriately and rigorously? 

Reviewer #1: N/A

Reviewer #2: N/A

3. Have the authors made all data underlying the findings in their manuscript fully available?

Reviewer #1: Yes

Reviewer #2: Yes

4. Is the manuscript presented in an intelligible fashion and written in standard English?

Reviewer #1: No

Reviewer #2: Yes

5. Review Comments to the Author

Reviewer #1: Summary

This is a useful review of machine learning methods for risk prediction of diseases or their complications using primary care databases. I’ve identified a number of issues with it, but, if these can be resolved, I would recommend publication.

Main comments

Many people will object (some strongly) to the inclusion of logistic regression here as an ML method. Some would also argue, for example, that ANNs can be just a set of logistic regressions and would therefore also be included in statistical learning rather than ML. “Classical log reg” is mentioned once, so it’s important to explain what is meant by this and by simply “log reg”.

On the same theme, I would not include the studies that used clustering or other unsupervised methods, as these are not really about risk prediction for an individual patient. If reinforcement and unsupervised ML methods, e.g., NLP have been used in pre-processing, they should be separated from the main method used to fit the model.

The tables are good. However, the accompanying text often repeats what’s in the tables in terms of how many studies were of which diagnosis or came from which country. I think this would be easier to digest if placed in an extra table instead.

Also, the text brings out some important points from a particular study but without developing it or framing it for the reader. For example, “a study reported that variations in the importance of different risk factors depend on the modelling technique [35]”. Was this the only study to do this? Is one of the goals of this paper under review to highlight such modelling issues or just to summarise the frequency of methods used? The two sentences that followed that one also illustrate the point: “Another study reported that ignoring censoring substantially underestimated risk of cardiovascular disease [43]. Also, systolic blood pressure could be used as a predictor of hospitalization or mortality [59].” It’s well known that censoring affects the estimated risk, so this sentence should be deleted. If SBP is worth mentioning, what about other predictors? There are many other such examples in other sections of the Results, which needs to be rethought.

Some reflection on WHY the reporting of methods in the studies in this review is often so limited would be useful. There have been many studies highlighting such limitations in non-ML fields that are relevant here.

Discussion: “Other drawbacks reported, similar to our findings, were underrepresentation of healthy persons and retrospective temporal dimension of the extracted predictors [142].” This is worth explaining better and unpacking with further discussion. For example, is the underrepresentation due to healthy people not attending PHC and so not being captured in the EHR? Or is something else meant here?

Discussion: “Furthermore, ML engineers must be aware of the unintended implications of their algorithms and ensure that they are created with the global and local communities in mind [145].” Again, this needs explaining and unpicking. What are the unintended implications? These could include healthcare rationing and widening of inequalities, but the authors may be thinking of something else (too). Do algorithms need to consider “global communities”? This brings me to a related point about the need for external validation of models. I would argue that external validation of a model in another country is not needed unless the model’s creators intend the model to be used overseas. Healthcare systems – and hence the populations using a particular sector and the available data – vary so much that performance would be either too poor to use or the model simply not relevant. The authors have penalised the lack of external validation perhaps unfairly. However, if what is meant by “external validation” covers another similar data set in the same country, then I’d agree that that validation would be important to do.

Minor comments

The Abstract should give the name of the risk of bias tool. “Extracted date” should be “Extracted data”. %s should be given for Alzheimer’s disease and diabetes. The Conclusions should summarise the importance / relevance of the findings rather than say it’s the first of its kind.

Introduction: “To achieve these PHC care aims, common health disorders require risk prediction for primary prevention, early diagnosis, follow-up, and timely interventions to avoid diseases exacerbations and complications.” I disagree that risk prediction is needed for all these things if it refers to risk-prediction models. Clinician training and experience is used far more than any model for diagnosis, for instance. The rationale for risk-prediction models needs to be clearly and accurately set out to help demonstrate the importance of this study.

Introduction: most EHR-type databases available to researchers are not “big data”, simply large. This is one reason why statistical learning is so common in risk-prediction models.

Methods: “Health conditions extracted were categorized according to international classification

of diseases (ICD)-10 version 2019”. How was this done for UK studies, where primary care EHRs don’t use ICD10?

Results: “Sample sizes used for training and/or validating the models across the included studies

ranged from 26 to around 4 million participants”. Is 26 right?!?

Results: “A study revealed that models can be created using only data from medical records and had prediction values of 70-80% for identifying persons who are at risk of acquiring ankylosing spondylitis (M45) [100].” Do you mean a sensitivity of 70-80%? Or PPV? Or something else?

Weng (ref 35) used CPRD, and therefore any models from it are by definition – in contrast to what Weng et al say in their paper – retrospective cohort studies. Table 1 should be amended accordingly and the other CPRD studies checked.

Discussion: “it would be advisable that models’ developers propose solutions for the digital documentation systems…” Modellers are not IT or IG experts!

There are numerous minor issues with the English. Just one example is: “A few studies (n=10) compared the performance of the developed ML models to other standard reference techniques”, where “to” should be “with” (they mean different things).

Another is: “Children obesity” should be “childhood obesity”.

“Evitable” should be “avoidable”.

There are many others.

Reviewer #2: Recommendation: minor revision

This work presents a systematic review of the application of ML in the context of primary health care. Literature published between January 1990 to January 2022 is considered.

1. Is the manuscript technically sound, and do the data support the conclusions? Yes

2. Has the statistical analysis been performed appropriately and rigorously? NA

3. Have the authors made all data underlying the findings in their manuscript fully available? NA

4. Is the manuscript presented in an intelligible fashion and written in standard English? Yes

1. No insights related to algorithms applied. Any observation related to the most commonly used algorithms and the best performing ones?

2. What kind of data is usually used? Textual? Images? What kind of features?

3. What ate the current challenges in applying ML to primary health care data?

4. What are the most challenging diseases for ML prediction?

5. What kind of evaluation approaches are used?

6. Has any of the reviewed work been deployed in a real world application?

7. Quality of figures and tables need to be improved.

8. Need to discuss implications of the results.

6. PLOS authors have the option to publish the peer review history of their article (what does this mean?). If published, this will include your full peer review and any attached files.

Reviewer #1: **Yes: **Alex Bottle

Reviewer #2: No

---

## [Author Response · Author response to Decision Letter 0]

25 May 2023

Response to reviewers' comments

Reviewer #1: 

Summary: This is a useful review of machine learning methods for risk prediction of diseases or their complications using primary care databases. I’ve identified a number of issues with it, but, if these can be resolved, I would recommend publication.

Thank you for your comments, we highly appreciate your valuable feedback.

Main comments

1. Many people will object (some strongly) to the inclusion of logistic regression here as an ML method. Some would also argue, for example, that ANNs can be just a set of logistic regressions and would therefore also be included in statistical learning rather than ML. “Classical log reg” is mentioned once, so it’s important to explain what is meant by this and by simply “log reg”.

We agree that logistic regression is not a method of machine learning but is rather counted under statistical learning. Therefore, we follow your suggestion and omitted logistic regression from the methods investigated. This did not affect the number of the included studies, however the total number of models was reduced (273 to 224 models). Subsequently, we removed this sentence “A few studies (n=10) compared the performance of the developed ML models to other standard reference techniques that were based on classical statistics, such as classical logistic and Cox regression.”

However, we prefer to keep ANNs. Even if it is true that they can be seen as a set of logistic regressions, ANNs usually develop more complex functional relationships between predictors and outcome than logistic regression and are typically counted as machine learning models. 

2. On the same theme, I would not include the studies that used clustering or other unsupervised methods, as these are not really about risk prediction for an individual patient. If reinforcement and unsupervised ML methods, e.g., NLP have been used in pre-processing, they should be separated from the main method used to fit the model.

Clustering is not for risk prediction for individual patients. The primary studies using unsupervised models were included to identify the health conditions addressed within primary healthcare settings. But considering your valuable comment, our results regarding the unsupervised ML models would not be reliable, as many similar studies might have been missed at the literature screening stage. Hence, these studies using only unsupervised ML models were removed from the included studies (n=3, namely Alexander et al 2021, Nikolaou et al 2021, and Pikoula et al 2019).

Regarding NLP, we removed it from the descriptive summary of the ML predictive models and rephrased the sentence to clarify its usage as a preparatory step. The sentence was corrected (Lines 205-207) as follows: “Five studies used the natural language processing (NLP) technique to handle free text clinical notes [43,45–48].”

3. The tables are good. However, the accompanying text often repeats what’s in the tables in terms of how many studies were of which diagnosis or came from which country. I think this would be easier to digest if placed in an extra table instead.

The captions of the three tables (Tables 2-4) were meant to highlight the grouping of the primary studies as indicated in the methods section. The additional accompanying descriptive text in the “Results” section was removed. An extra panel was provided as a supplementary material to quantitatively summarize the results (S1 Panel). 

4. Also, the text brings out some important points from a particular study but without developing it or framing it for the reader. For example, “a study reported that variations in the importance of different risk factors depend on the modelling technique [35]”. Was this the only study to do this? Is one of the goals of this paper under review to highlight such modelling issues or just to summarise the frequency of methods used? The two sentences that followed that one also illustrate the point: “Another study reported that ignoring censoring substantially underestimated risk of cardiovascular disease [43]. Also, systolic blood pressure could be used as a predictor of hospitalization or mortality [59].” It’s well known that censoring affects the estimated risk, so this sentence should be deleted. If SBP is worth mentioning, what about other predictors? There are many other such examples in other sections of the Results, which needs to be rethought.

In our protocol, we planned to encompass the health conditions as well as a deeper statistical analysis of performance of the models tested and the used predictors in two manuscripts. Considering that the points you highlighted could have been quite selective and biased, we removed all these selected points. Hence, we keep this manuscript consistent with one clear outcome and maintain it as an umbrella presentation of the health conditions. Therefore, we updated all the relevant points across the manuscript accordingly. 

5. Some reflection on WHY the reporting of methods in the studies in this review is often so limited would be useful. There have been many studies highlighting such limitations in non-ML fields that are relevant here.

We highlighted this point in the “discussion” sections, lines 355-357 as follows: “Navarro and colleagues investigated this underreporting, and they claimed that the available reporting guidelines of modelling studies might be less apposite for ML models studies [141]”

6. Discussion: “Other drawbacks reported, similar to our findings, were underrepresentation of healthy persons and retrospective temporal dimension of the extracted predictors [142].” This is worth explaining better and unpacking with further discussion. For example, is the underrepresentation due to healthy people not attending PHC and so not being captured in the EHR? Or is something else meant here?

The discussion section was re-written and reorganized for better interpretation. This point is interpreted in the “Implication of results” subsection, lines 371-375 as follows: “Well-designed and -documented repositories provide representative health data for the healthy and diseased populations are needed [140]. These data repositories might support matching the research questions of future modelling studies with the models’ development and validation process for more accurate prediction.” 

7. Discussion: “Furthermore, ML engineers must be aware of the unintended implications of their algorithms and ensure that they are created with the global and local communities in mind [145].” Again, this needs explaining and unpicking. What are the unintended implications? These could include healthcare rationing and widening of inequalities, but the authors may be thinking of something else (too). Do algorithms need to consider “global communities”?

This point was also rephrased (lines 362-369). To avoid unclarity, we omitted “global and local communities.” We wrote: “Interdisciplinary collaboration among healthcare workers, developers of ML models, and creators of digital documentation systems is required. This is especially important given the increasing popularity of digitally connected health systems [5]. It is recommended to augment the participation of health professionals through the development process of the PHC predictive models to critically evaluate, assess, adopt, and challenge the validation of the models within practices. This collaboration may assist ML engineers to recognize unintended negative implications of their algorithms, such as accidentally fitting of confounders, and unintended discriminatory bias, among others, for better health outcomes [142]”

8. This brings me to a related point about the need for external validation of models. I would argue that external validation of a model in another country is not needed unless the model’s creators intend the model to be used overseas. Healthcare systems – and hence the populations using a particular sector and the available data – vary so much that performance would be either too poor to use or the model simply not relevant. The authors have penalised the lack of external validation perhaps unfairly. However, if what is meant by “external validation” covers another similar data set in the same country, then I’d agree that that validation would be important to do.

Lack of external validation was only considered based on the CHARMS and PROBAST guidelines in the case of complete absence of validation of the model using different dataset. Those studies reporting validation of the models from different times in the same source or different datasets within the same country were considered externally validated models. To support reading clarity, this sentence was added in the “Methods” section, lines 138-141 : “External validation was considered if the model was at least validated using a dataset from a later time point in the same data source (temporal external validation) or using a different dataset from inside or outside the source country (geographical or broad external validation, respectively) [20].”

9. Minor comments

a. The Abstract should give the name of the risk of bias tool. “Extracted date” should be “Extracted data”. %s should be given for Alzheimer’s disease and diabetes.

Corrected

b. The Conclusions should summarise the importance / relevance of the findings rather than say it’s the first of its kind.

We updated this sentence to the following: “Our study provides a summary on the presently available ML prediction models within PHC. We draw the attention of digital health policy makers, ML models developer, and healthcare professionals for more future interdisciplinary research collaboration in this regard.” 

c. Introduction: “To achieve these PHC care aims, common health disorders require risk prediction for primary prevention, early diagnosis, follow-up, and timely interventions to avoid diseases exacerbations and complications.” I disagree that risk prediction is needed for all these things if it refers to risk-prediction models. Clinician training and experience is used far more than any model for diagnosis, for instance. The rationale for risk-prediction models needs to be clearly and accurately set out to help demonstrate the importance of this study.

We agree with your comment. We rephrased it for more clarity (Lines 46-51) as follows: “This form of health maintenance, which aims to provide constant access to high-quality care and comprehensive services, is defined and called for by the World Health Organization (WHO) global vision for PHC [2]. Clinicians’ skills and experience and the further continuing professional development are fundamental to achieve these PHC care aims [3]. Additional health care improvement can be achieved by capitalizing on digital health and AI technologies.”

d. Introduction: most EHR-type databases available to researchers are not “big data”, simply large. This is one reason why statistical learning is so common in risk-prediction models.

We rephrased this sentence and omitted the term “Big Data” and rephrased it as follows: “With the high number of patients visiting PHC and the emergence of electronic health records, substantial amounts of data are generated on daily basis.”

e. Methods: “Health conditions extracted were categorized according to international classification of diseases (ICD)-10 version 2019”. How was this done for UK studies, where primary care EHRs don’t use ICD10?

We updated the explanation of this point in the “Methods” section, Lines 119-122, as follows: “This coding system was selected because it is applied by at least 120 countries across the globe [24]. Considering the countries that apply different coding systems, we used the explicit names of the health condition mentioned in the primary studies included to match them to the closest ICD-10 codes.”

f. Results: “Sample sizes used for training and/or validating the models across the included studies ranged from 26 to around 4 million participants”. Is 26 right?!?

This study (Levy et al 2018) has the lowest populations. The training data were from 49 populations while the external validation data were only 26. We updated it to the total populations of the study (75) in line 69. 

g. “A study revealed that models can be created using only data from medical records and had prediction values of 70-80% for identifying persons who are at risk of acquiring ankylosing spondylitis (M45) [100].” Do you mean a sensitivity of 70-80%? Or PPV? Or something else?

This sentence was deleted for the reason mentioned in response to point number 4.

h. Weng (ref 35) used CPRD, and therefore any models from it are by definition – in contrast to what Weng et al say in their paper – retrospective cohort studies. Table 1 should be amended accordingly and the other CPRD studies checked.

All the studies rechecked and updated accordingly in the tables and the relevant paragraphs.

i. Discussion: 

The discussion section was re-written and reorganized for better interpretation. 

j. “it would be advisable that models’ developers propose solutions for the digital documentation systems…” Modellers are not IT or IG experts!

The sentence was unclear and we omitted it.

k. There are numerous minor issues with the English. Just one example is: “A few studies (n=10) compared the performance of the developed ML models to other standard reference techniques”, where “to” should be “with” (they mean different things). Another is: “Children obesity” should be “childhood obesity”., “Evitable” should be “avoidable”. There are many others.

Corrected. The native English author SW. applied further language improvement.

Reviewer #2: 

Recommendation: minor revision

This work presents a systematic review of the application of ML in the context of primary health care. Literature published between January 1990 to January 2022 is considered.

1. No insights related to algorithms applied. Any observation related to the most commonly used algorithms and the best performing ones?

As this manuscript’s main objective was to highlight the health conditions targeted by the primary studies, we did not present the statistical estimation of the model’s performance for the reported models. However, we had already extracted such results to be presented with deeper analysis in a more homogenous and cohesive manner in another manuscript for some health conditions.

2. What kind of data is usually used? Textual? Images? What kind of features?

We added a paragraph to briefly highlight the categories features/ predictors we found, lines 200-207, as follows: “A variety of predictors were described across the studies. Demographic characteristics and clinical picture of the health conditions were the most frequently found predictors. Medications, comorbidities, and blood tests performed within primary care unit were reported. Predictors, such as blood test results and imaging results performed within secondary and tertiary health care were additionally reported in some of the individual studies. Referral documentation and clinical notes taken by health care personnel were also reported. Five studies used the natural language processing (NLP) technique to handle free text clinical notes [43,45–48].”

3. What ate the current challenges in applying ML to primary health care data?

The “Discussion” section was re-written and reorganized to highlight the current challenges that we identified. 

4. What are the most challenging diseases for ML prediction?

To present such a result, we planned, according to our protocol, to present a deeper analysis of the models and the predictors, tackling the challenges faced by the developers as well as their final conclusions regarding their models. This will be in the subsequent manuscript for a cohesive and consistent reporting of the results according to the ICD-10 category.

5. What kind of evaluation approaches are used?

As this manuscript’s main objective is to highlight the health conditions targeted by the primary studies, we did not report the results of the evaluation approaches. To avoid selective and biased results reporting as well, we removed the relevant text. Please see the answer of comment number 4 in the comments of reviewer 1.

6. Has any of the reviewed work been deployed in a real world application?

According to our knowledge and analysis of the full text of the included studies, only four studies claimed a real-time application. This was updated in lines 189-191. 

7. Quality of figures and tables need to be improved.

Done

8. Need to discuss implications of the results.

The discussion section was re-written and reorganized to highlight the implication of results, lines 360-385.

---

## [Decision Letter · Decision Letter 1]

19 Jun 2023

PONE-D-22-23382R1A systematic review of clinical health conditions predicted by machine learning diagnostic and prognostic models trained or validated using real-world primary health care dataPLOS ONE

Dear Dr. Abdulazeem,

Thank you for submitting your manuscript to PLOS ONE. After careful consideration, we feel that it has merit but does not fully meet PLOS ONE’s publication criteria as it currently stands. Therefore, we invite you to submit a revised version of the manuscript that addresses the points raised during the review process.

We look forward to receiving your revised manuscript.

Kind regards,

Ágnes Vathy-Fogarassy, Ph.D.

Academic Editor

PLOS ONE

Additional Editor Comments:

Dear Authors,

For the corrected manuscript, Reviewer #2 gave the following feedback:

"Some of my concerns were not addressed in your response."

Please take into account all the reviewer's suggestions and requests, otherwise we cannot accept the article.

Best regards!

Reviewers' comments:

Reviewer's Responses to Questions

**Comments to the Author**

1. If the authors have adequately addressed your comments raised in a previous round of review and you feel that this manuscript is now acceptable for publication, you may indicate that here to bypass the “Comments to the Author” section, enter your conflict of interest statement in the “Confidential to Editor” section, and submit your "Accept" recommendation.

Reviewer #1: All comments have been addressed

Reviewer #2: (No Response)

2. Is the manuscript technically sound, and do the data support the conclusions?

Reviewer #1: Yes

Reviewer #2: Partly

3. Has the statistical analysis been performed appropriately and rigorously? 

Reviewer #1: Yes

Reviewer #2: N/A

4. Have the authors made all data underlying the findings in their manuscript fully available?

Reviewer #1: Yes

Reviewer #2: Yes

5. Is the manuscript presented in an intelligible fashion and written in standard English?

Reviewer #1: Yes

Reviewer #2: Yes

6. Review Comments to the Author

Reviewer #1: The authors have responded well to my comments, and in my opinion the ms is much improved. I am happy to recommend it for publication.

Reviewer #2: (No Response)

7. PLOS authors have the option to publish the peer review history of their article (what does this mean?). If published, this will include your full peer review and any attached files.

Reviewer #1: No

Reviewer #2: No

---

## [Author Response · Author response to Decision Letter 1]

21 Aug 2023

Revision 2: Response to Reviewers' Comments

Response to Reviewer #2: 

We would like to thank you for your valuable comments that helped us improve our manuscript. We are very eager to properly address all your comments. Despite that, it seems that we did not address some of your comments. We apologize for this. 

We now addressed all comments in detail, restructured parts of the discussion and added a new table (S2 Table) to the supplement addressing the characteristics of all ML predictive models included in the manuscript (see below).

1. No insights related to algorithms applied. Any observation related to the most commonly used algorithms and the best performing ones?

We addressed this point in our manuscript through presenting the advantages and disadvantages of the included models and related that to our results across the manuscript:

o We referred to it in the introduction section in page 3, lines 60-61 as follows “The variety of ML prediction models’ characteristics provide broader opportunities to support the healthcare practice.”

o We presented more information regarding the advantages and disadvantages of the models included in a new supplementary Table S2. We referred to it in our results section page 10, lines 200-201 as follows: “Supplementary table (S2 Table) presents advantages and disadvantages of these models in addition to further descriptive information of the included studies.”

o In the discussion section, we discussed the most commonly used models in page 20, line 323-325 as follows: “In 106 observational studies, we identified 42 health conditions targeted by 207 ML prediction models, of which 38% were random forest and SVM. The included models used PHC data documented over the past 40 years for a total of 22.7 million patients.”. Please also see answer to question 2 below.

o Furthermore, we related our results to the advantages of these models and the potential benefit of it within the primary healthcare and referred to it in page 22, lines 360-369 as follows: “Models that are known for their high prediction accuracy, such as random forest and SVM might support better health outcomes when developed using high quality health data [140]. Additionally, the variety of the ML prediction models characteristics provide opportunities to improve healthcare practice. Using large data documented as electronic health records, random forest models and ensemble models such as boosting models have the ability to handle large datasets with numerous predictors variables [141]. Artificial neural network can also perform complex images processing that can boost the primary health care services [141]. Furthermore, SVM and decision tree models can provide nonlinear solutions, thus will support our understanding of complex and dynamic diseases for earlier health conditions prediction [142].” 

2. What kind of data is usually used? Textual? Images? What kind of features?

 We added a paragraph to highlight the data used in page 10, lines 207-2014, as follows: “The data used to develop the models were called predictors, features, or variables across the included studies. These data were mostly textual. Demographic characteristics and clinical picture of the health conditions were the most frequently found data. Medications, comorbidities, and blood tests performed within primary care unit were reported. Data, such as blood test results and imaging results performed within secondary and tertiary health care were additionally reported in some of the individual studies. Referral documentation and clinical notes taken by health care personnel were also reported. Five studies used the natural language processing (NLP) technique to handle free text clinical notes [43,45–48].”

3. What are the current challenges in applying ML to primary health care data?

We highlighted some of the current challenges in applying ML to PHC data in the discussion section and presented it in context with our results in the subsection “Results in perspective” in two places:

o in page 21, lines 344-352 as follows: “The coding system used in health records does not universally follow the same criteria for all diseases, posing challenges for the consistency of models’ performance [137]. Moreover, the lack of globally standardized definitions and terminology of diseases and the wide variability of the services provided across different health systems further limit the effectiveness of the models [137]. For example, uncoded free-text clinical notes as well as using ‘race’ and ‘ethnicity’ or ‘suicide’ and ‘suicide attempts’ to be documented as a single input can affect the predictive power of the models [138]. Other drawbacks reported include underrepresentation of healthy persons, retrospective temporal dimension of predictors, and the absence of confirmatory diagnostic services in PHC pose significant limitations [140, 14 1].” 

o in page 22, lines 378-386, as follows “Health care professionals are fundamental to the process of implementing and integrating ML prediction models in their healthcare practice. Despite that, our review did not report outcomes related to healthcare professionals. Significant variability of opinions on the utilization of ML in PHC among primary health care providers hinder its acceptance. Furthermore, the black-box nature of ML prediction models precludes the clinical interpretability of models’ outcomes. Additional workload and training are needed to implement such technology in the routine practice. Trust, data protection, and ethical and clinical responsibility legislation are further intractable issues that represent major obstacles toward ML prediction models implementation [5].”

o In page 24, lines 410-413, as follows “Despite the additional burden, it is beneficial also to continuously assess the potential significance of models, such as improved health outcomes, reduced medical errors, increased professional effectiveness and productivity, and enhanced patients’ quality of life [147].”

4. What are the most challenging diseases for ML prediction?

We addressed this point in page 22, lines “370-377” as follows: “Nature of diseases append further challenges. The most challenging diseases for ML prediction are multifaceted long-term health conditions, such as DM, that are influenced by combination of genetic, environmental, and lifestyle factors. The complex health conditions further tangle the models, making it harder to identify accurate predictive patterns. Furthermore, the subjective nature of symptoms, especially symptoms related to mental health disorders, pose additive challenges toward ML models accuracy. Rare diseases, if documented, often suffer from limited data availability, leading to difficulty to train ML models effectively [143].”

5. What kind of evaluation approaches are used?

We addressed this point in page 10, lines 201-206 as follows: “The most frequently reported evaluation approach of models’ performance was the area under the receiver operating characteristic curve (AUROC), which was reported as “good” to “moderate” models performance in 62 studies. One study reported the performance measures using decision analysis curve [32]. Other evaluation approaches were reported across the included studies, such as calculating sensitivity, specificity, predictive values, and accuracy.”

6. Has any of the reviewed work been deployed in a real world application?

We referred to this point in page 9, lines 190-192 “Despite all studies included used real-world patients' data to develop and/or validate the ML models, four studies reported applying the models develop in real-world primary health care settings [127,128, 130, 131].” 

7. Quality of figures and tables need to be improved.

We improved the quality of all our tables and figure by proper reformatting for font, spacing, and colours.

8. Need to discuss implications of the results.

The discussion section was re-written and reorganized to highlight the implication of results. We added a subsection in pages 23-24, lines 393-422 as follows “Implication of results and recommendation for future contributions: This review provided a comprehensive outline of ML prediction models in PHC and raises important considerations for future research and implementation of this technology in PHC settings. Interdisciplinary collaboration among health care workers, developers of ML models, and creators of digital documentation systems is required. This is especially important given the increasing popularity of digitally connected health systems [5]. It is recommended to augment the participation of health professionals through the development process of the PHC predictive models to critically evaluate, assess, adopt, and challenge the validation of the models within practices. This collaboration may assist ML engineers to recognize unintended negative implications of their algorithms, such as accidentally fitting of confounders, and unintended discriminatory bias, among others, for better health outcomes [146]. Health care systems need to provide comprehensive population health data repositories as an enabler for medical analyses [137]. Well-designed and -documented repositories which provide representative health data for the healthy and diseased populations are needed [137, 140]. These high quality data repositories might provide future modelling studies with data that match the studies’ clinical research questions for more accurate prediction. Further ML prediction studies are needed to target more health conditions using PHC data. Despite being challenging, it is beneficial to early assess the potential significance of models, such as improved health outcomes, reduced medical errors, increased professional effectiveness and productivity, and enhanced patients’ quality of life [147]. It is recommended to follow reporting guidelines for producing valid and reproducible ML modelling studies. Developing robust frameworks to enable the adoption and integration of ML models in the routine practice is also essential for effective transition from conventional health care systems to digital health [148,149]. Sophisticated technical infrastructure and strong academic and governmental support are essential for promoting and supporting long-term and broad-reaching PHC ML-based services [138,150]. However, balanced arguments [151,152] regarding the potential benefits and limitations of ML models support better health care without overestimating or hampering the use of such technology. It is also suggested to integrate the basic understanding of ML concepts and techniques in education programs for health science and medical students.”

---

## [Editor Report · Decision Letter 2]

29 Aug 2023

A systematic review of clinical health conditions predicted by machine learning diagnostic and prognostic models trained or validated using real-world primary health care data

PONE-D-22-23382R2

Dear Dr. Abdulazeem,

We’re pleased to inform you that your manuscript has been judged scientifically suitable for publication and will be formally accepted for publication once it meets all outstanding technical requirements.

Kind regards,

Ágnes Vathy-Fogarassy, Ph.D.

Academic Editor

PLOS ONE

---

## [Editor Report · Acceptance letter]

31 Aug 2023

PONE-D-22-23382R2 

A systematic review of clinical health conditions predicted by machine learning diagnostic and prognostic models trained or validated using real-world primary health care data 

Dear Dr. Abdulazeem:

I'm pleased to inform you that your manuscript has been deemed suitable for publication in PLOS ONE. Congratulations! Your manuscript is now with our production department. 

Kind regards, 

on behalf of

Dr. Ágnes Vathy-Fogarassy 

Academic Editor

PLOS ONE